# AI-guided patient stratification improves outcomes and efficiency in the AMARANTH Alzheimer's Disease clinical trial

Delshad Vaghari[1,6], Gayathri Mohankumar[2,6], Keith Tan[3], Andrew Lowe[3], Craig Shering[4], Peter Tino[5] & Zoe Kourtzi [1] ✉

Alzheimer's Disease (AD) drug discovery has been hampered by patient heterogeneity, and the lack of sensitive tools for precise stratification. Here, we demonstrate that our robust and interpretable AI-guided tool (predictive prognostic model, PPM) enhances precision in patient stratification, improving outcomes and decreasing sample size for a AD clinical trial. The AMARANTH trial of lanabecestat, a BACE1 inhibitor, was deemed futile, as treatment did not change cognitive outcomes, despite reducing β-amyloid. Employing the PPM, we re-stratify patients precisely using baseline data and demonstrate significant treatment effects; that is, 46% slowing of cognitive decline for slow progressive patients at earlier stages of neurodegeneration. In contrast, rapid progressive patients did not show significant change in cognitive outcomes. Our results provide evidence for AI-guided patient stratification that is more precise than standard patient selection approaches (e.g. β-amyloid positivity) and has strong potential to enhance efficiency and efficacy of future AD trials.

Dementia presents a major global healthcare challenge, affecting more than 55 million individuals around the world, with a projected three-fold increase over the next 50 years[1]. Alzheimer's disease (AD), the predominant cause of dementia, accounts for 60–80% of dementia cases[2]. Effective treatments to prevent the onset, delay the progression, or modify the course of AD are urgently needed to reduce the global burden[3]. Despite decades of research and development, clinical trials of potential disease-modifying treatments for dementia have been largely unsuccessful. Cumulative expenditure on clinical-stage AD research and development is estimated to have reached $42.5 billion since 1995[4,5], and the cost to develop a treatment for AD from the preclinical stage to FDA approval is estimated to be $5.7 billion[6]. Recruitment and patient selection from diverse and qualified pools of volunteers[7,8] often cause significant delays and contribute to the high failure rates of these trials[4,5].

Recent positive phase three clinical trial results (i.e., lecanemab, donanemab)[9,10] highlight the need for interventions earlier in the progression of disease when treatments may be maximally effective[11,12] and have the potential to enhance patient outcomes. Yet, we still lack effective tools for precise stratification of patients at risk or early disease stages for inclusion in clinical trials. In particular, patient selection often relies on biomarkers (e.g., β-Amyloid) that are limited in predicting AD progression and treatment outcomes due to variability in amyloid binding and immune activation[3,4,13,14]. Further, up to a third of patients at early Mild Cognitive Impairment (MCI) stages may be misdiagnosed due to a lack of sensitive tools for early diagnosis. Including patients with symptoms due to comorbidities (e.g., anxiety or mood-related disorders) rather than dementia pathology in clinical trials may impact trial efficiency and costs (i.e., larger numbers of patients and longer recruitment are necessary), as well as confound

[1]Department of Psychology, University of Cambridge, Cambridge, UK. [2]Centre for AI, Data Science & Artificial Intelligence, BioPharmaceuticals R&D, AstraZeneca, Gaithersburg, USA. [3]Neuroscience, BioPharmaceuticals R&D, AstraZeneca, Cambridge, UK. [4]Neuroscience, BioPharmaceuticals R&D, AstraZeneca, Boston, USA. [5]School of Computer Science, University of Birmingham, Birmingham, United Kingdom. [6]These authors contributed equally: Delshad Vaghari, Gayathri Mohankumar. ✉e-mail: zk240@cam.ac.uk

efficacy signals due to increased heterogeneity in the patient sample. Further, including patients who have progressed to advanced disease stages may reduce the potential of targets (e.g., amyloid removal targets) to be effective[3,15–17].

Recent developments in Artificial Intelligence (AI) based on machine learning (ML) algorithms provide a turning point in precise patient stratification in early dementia stages. We have developed and validated a robust and interpretable predictive prognostic model (PPM) that extends beyond binary patient classification approaches and predicts not only whether but also how fast individuals at early stages of the disease (MCI) or even pre-symptomatic stages (Cognitive Normal, CN) may progress to AD[18,19]. PPM has been tested on independent real-world patient data from memory clinics and validated against longitudinal clinical outcomes[20]. PPM delivers an AI-guided marker of future cognitive health that predicts progression to AD more precisely than standard clinical assessments (i.e., cognitive data, MRI scan)[20] and biomarkers typically used for patient inclusion in clinical trials (i.e., β-Amyloid positivity)[19], offering potential to reduce misdiagnosis and optimize patient stratification.

Here, we test whether employing the PPM to improve patient stratification may change the outcome and efficiency of a randomized phase 2/3 clinical trial (AMARANTH, NCT02245737). AMARANTH tested the effect of lanabecestat (AZD3293, LY3314814), a brain-permeable inhibitor of human beta-site amyloid precursor protein-cleaving enzyme 1 (BACE1/β-secretase)[21]. BACE1 was considered to be a promising therapeutic target for slowing disease progression in AD by preventing the generation of Aβ peptides, reducing the effects of Aβ toxicity and the formation of amyloid plaques in the brain. The objective of the trial was to test the efficacy of lanabecestat in slowing cognitive decline in patients diagnosed with MCI due to AD or mild AD. Despite lanabecestat reducing β-amyloid, the readout of the trial was deemed unsuccessful, and the trial was terminated early due to a lack of significant changes in primary cognitive outcomes.

We employed the PPM– trained on research data (ADNI) –to re-stratify individuals in the AMARANTH trial data (independent test data) into slow vs. rapid progressive based on baseline data (i.e., week 1, before treatment). In contrast to the futility assessment, we demonstrate significant treatment effects on primary trial outcomes. In particular, we test the effect of lanabecestat (20 mg vs. 50 mg) on outcomes (β-amyloid, cognition: CDR-SOB, ADAS-Cog13) separately for slow vs. rapid progressive individuals. Our results demonstrate 46% slowing of cognitive decline (as measured by CDR-SOB) for the slow rather than the rapid progressive group following treatment with lanabecestat 50 mg compared to placebo. That is, patients stratified by the PPM as slow progressive at earlier stages of neurodegeneration showed slowing of cognitive decline related to β-amyloid reduction due to treatment. Further, we show that using PPM for patient stratification substantially decreases the sample size necessary for identifying significant changes in cognitive outcomes. Our results suggest that using PPM to stratify patients for clinical trials has strong potential to enhance their efficiency (faster and cheaper) and efficacy (more reliable outcomes), as the right patients are included in the trials at the right time.

## Results

### PPM trained on ADNI stratifies clinically stable vs. declining individuals

PPM adopts a trajectory modeling approach based on Generalized Metric Learning Vector Quantization (GMLVQ)[18,19] that leverages multimodal data to make predictions about future cognitive decline at early dementia stages by iteratively adjusting class-specific prototypes and learning class boundaries (Supplementary Material: Predictive prognostic model). GMLVQ incorporates a full metric tensor to provide a robust distance measure (metric) tuned to the classification task. We trained the PPM on baseline data from ADNI ($n = 256$) to discriminate Clinically Stable ($n = 100$) from Clinically Declining ($n = 156$) patients, using β-Amyloid, APOE4, and medial temporal lobe (MTL) GM density. Employing ensemble learning with cross-validation and majority voting showed 91.1% classification accuracy (0.94 AUC: Area Under Curve) with sensitivity of 87.5% and specificity of 94.2% (Supplementary Table S1; Precision: 93.8%, F1 score: 90.5%). Note that discriminating Clinically Stable vs. declining individuals comprises a finer classification task compared to previous work focusing on patient (AD) vs. healthy (cognitive normal) classifications[22,23], where signals are more discriminable and may result in higher model performance. The difference between sensitivity and specificity is likely due to weaker signals for the clinically declining compared to the stable class. Thus, achieving higher than 90% accuracy and precision provides evidence for PPM robustness.

Further, the PPM architecture is transparent and interpretable. First, interrogating the metric tensors allows us to understand the contribution of each feature to the model's prediction (Fig. 1A). In particular, the metric tensors indicate that β-amyloid burden is the most discriminative feature compared to MTL GM density and APOE4. This is consistent with the role of β-amyloid and MTL atrophy as markers of Alzheimer's pathology, consistent with the NIA-AA 2018 diagnostic framework of AD[24]

Second, interrogating the off-diagonal terms of the metric tensor allows us to understand the feature interactions that contribute to the model's prediction. In particular, we observed a positive interaction between baseline β-amyloid burden and APOE 4, while a negative interaction between baseline β-amyloid burden and MTL GM density, consistent with the role of β-amyloid and APOE 4 as risk factors for progression to AD.

Third, the PPM prototypes (one per class: clinically stable, progressive) indicate the most discriminative class representatives and allow us to predict an individual's trajectory. That is, using the GMLVQ-Scalar Projection method, we estimate the distance (based on the learnt metric tensor) of an independent test dataset from the Clinically Stable prototype and determine the PPM-derived prognostic index for each individual, allowing individualized prognosis beyond binary clinical labels (Fig. 1B; Supplementary Material: GMLVQ-Scalar Projection). In particular, we extracted the PPM-derived prognostic index for each individual in an independent ADNI sample (out-of-sample validation, $n = 419$; cognitive normal individuals n = 119, patients with MCI $n = 150$, patients with AD $n = 150$). We then used multinomial logistic regression to capture the relationship of the PPM-derived prognostic index to the rate of future tau accumulation and determine boundaries for quartile classes that differ in likelihood of disease progression. We scaled the boundaries so that PPM-derived prognostic index indicates individuals who are more likely to: (1) remain stable (PPM index values below 0 fall within the 20th percentile of the future tau accumulation slope), (2) experience rapid progression (PPM index values higher than 1 fall above the 60th percentile of future tau accumulation), (3) experience slower progression (PPM index between 0 and 1)[18,19]. Our results (Fig. 1B) showed that the PPM-derived prognostic index was significantly different across groups (Kruskal-Wallis H test χ(2) = 121.46, $p < 0.001$), with a significantly higher index (Bonferroni corrected) for AD vs. MCI and CN ($p < 0.001$), MCI vs. CN ($p < 0.001$). This validation against clinical outcomes (i.e., diagnosis) provides evidence that the PPM-derived prognostic score is clinically relevant.

### PPM-guided stratification in the AMARANTH trial using baseline data

We used the PPM trained on ADNI data to extract the PPM-derived prognostic index from AMARANTH patients (Supplementary Tables S2, S3 for sample sizes) with: APOE4 at week 1, structural MRI (MTL GM density), florbetapir PET (β-amyloid) scans at three time points (week 1, 52, 104), and cognitive measures (CDR-SOB, ADAS-Cog13) at three time points (week 1, 52, 104). In particular, using the

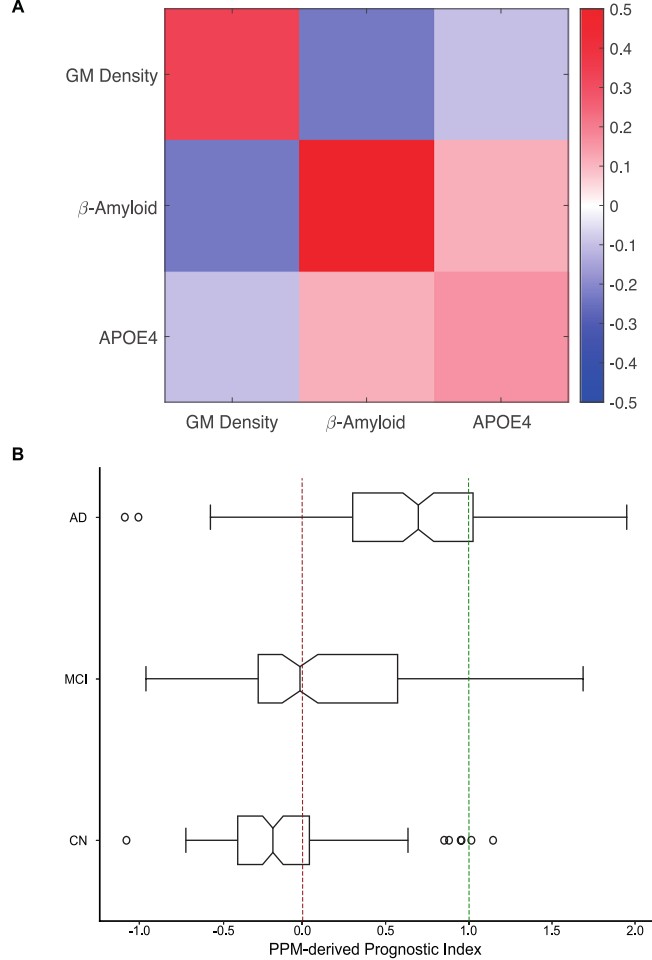

**Fig. 1 | PPM trained on ADNI data (*n* = 256) classifies Clinically Stable vs. Clinically declining individuals. A** PPM metric tensor based on training the PPM on MTL Gray Matter (GM) Density, β-Amyloid, and APOE 4 for model training. The color scale represents values for each cell in the metric tensor, with diagonal terms summing to 1. The diagonal terms show strong contribution of β-amyloid burden (weight: 0.51) compared to Gray matter density (weight: 0.34) and APOE 4 (weight: 0.15). **B** PPM-derived prognostic index for an independent ADNI validation dataset (*n* = 419): Box plots of PPM-derived prognostic index showing significant differences between Cognitive Normal: CN, Mild Cognitive Impairment: MCI, Alzheimer's Disease: AD (Kruskal-Wallis H two-sided tests, *p* < 0.001, Bonferroni corrected). Notches in the box plots indicate the median, the solid black box represents the 25th to 75th percentile, the black horizontal lines represent the range of the data, black circles indicate outliers, and non-overlapping notches indicate significantly different medians (*p* < 0.05). PPM-derived prognostic index below 0 indicates stable, above 1 indicates rapid progressive, and between 0 and 1 indicates slow progressive individuals. Dashed lines indicate boundaries between stable vs. slow progressive (red) and rapid progressive (green) based on a multinomial logistic regression testing the relationship of the PPM-derived prognostic index to the rate of future tau accumulation. Source data are provided as a Source Data file.

GMLVQ-Scalar Projection method, we estimated the distance of each patient in the AMARANTH dataset at baseline (week 1) from the Clinically Stable prototype, determined the PPM-derived prognostic index for each individual in the AMARANTH sample and stratified individuals as slow vs. rapid progressive based on baseline (week 1) data (Fig. 2A; the sample size for stable was small (*n* = 5) and these data were excluded from further analysis). There were no significant differences in the PPM-derived prognostic index between treatment groups at baseline (Kruskal-Wallis H test χ(2) = 2.9733, *p* = 0.2261; Fig. 2B). Interestingly, individuals in the slow progressive group showed lower β-amyloid burden (t (330.22) = 11.833, *p* < 0.001, higher

MTL GM density (t (326.33) = −17.351, *p* < 0.001) and better performance in cognitive tests (i.e., lower CDR-SOB (t(911.03) = 6.38, *p* < 0.001) and ADAS-Cog13 (t (806.77) = 5.23, *p* < 0.001)) compared to the rapid progressive group at baseline. These results suggest that individuals stratified as slow progressive by the PPM were at earlier stages of neurodegeneration and cognitive decline compared to individuals stratified as rapid progressive, corroborating the link of the PPM-derived prognostic score to cognitive decline and disease progression. Note that only 7.5% of the patients included in the trial had β-amyloid less than 50 centiloid, indicating intermediate or high likelihood of AD, and making it harder to stratify into subgroups based on β-amyloid alone. However, training the PPM on multimodal data (rather than β-amyloid alone) allowed a more precise stratification to subgroups at earlier vs. later disease progression stages.

## PPM-guided stratification in the AMARANTH trial shows treatment effects on β-Amyloid

We tested whether stratifying the AMARANTH dataset based on the PPM-derived index at baseline (week 1) shows treatment effects on β-Amyloid. In particular, we used a mixed model for repeated measures (MMRM; Supplementary Table S4), to test Treatment (placebo, 20 mg, 50 mg) effects across timepoints (week 1, 52, 104) for each PPM-stratified group (Slow vs. Rapid progressive).

We observed (Fig. 3A) a significant decrease in β-Amyloid for lanabecestat 20 mg and lanabecestat 50 mg treatment compared to placebo over time (week 104 compared to week 1). This decrease in β-Amyloid was observed for both PPM-stratified groups (Slow, Rapid progressive). In particular, MMRM analysis including fixed effects for treatment, timepoint, PPM-stratified group showed significant main effect of Timepoint (F(2, 485.43) = 37.01, *p* < 0.001) and PPM-stratified group (F(1, 489.19) = 84.04, *p* < 0.001), significant interactions for Treatment × Timepoint (F(4, 484.87) = 10.27, *p* < 0.001) and PPM-stratified group x Timepoint (F(2, 485.46) = 8.38, *p* < 0.001). Post-hoc comparisons showed that this PPM-stratified group x Timepoint interaction was significant for the lanabecestat 20 mg (F(2, 184.22) = 5.36, *p* < 0.01) and lanabecestat 50 mg (F(2, 146.80) = 3.47, *p* < 0.05) but not the placebo (F(2,152.76) = 1.20, *p* = 0.3033) group.

Further, computing change in β-Amyloid burden over time (week 104 minus week 1) corroborated these results showing significantly higher reduction in β-Amyloid for lanabecestat 20 mg and lanabecestat 50 mg compared to placebo for both the slow and rapid progressive group (Fig. 3B). In particular, we observed significantly higher reduction of β-Amyloid for a) lanabecestat 20 mg vs. placebo (Welch's Two Sample *t* test: Slow progressive group: t(55.289) = 2.95, *p* < 0.001; Rapid progressive group: t(154.29) = 4.10, *p* < 0 0.001) b) lanabecestat 50 mg vs. placebo (Welch's Two Sample *t* test: Slow progressive group: t(62.362) = 3.86, *p* < 0.001, Rapid progressive group: t(112.53) = 5.00, *p* < 0.001). Further, we observed stronger reduction in β-Amyloid due to treatment for the rapid (21.91 % change) than slow (17.14 % change) progressive group (Welch's Two Sample *t* test: lanabecestat 20 mg: t(73.062) = 3.31, *p* < 0.01; lanabecestat 50 mg: t(94.62) = 2.54, *p* = 0.013). This result was potentially due to the higher β-Amyloid burden for the rapid compared to the slow progressive group at baseline (week 1; t(330.22) = 11.83, *p* < 0.001).

Finally, similar analyses in the All Progressive group (*n* = 434, including individuals from the slow and rapid progressive groups) showed a significant reduction in β-Amyloid due to treatment over time compared to the placebo group. In particular, MMRM analysis showed a significant main effect of timepoint (F(2,482.33) = 55.65, *p* < 0.001) and significant interactions of treatment x timepoint (F(4,481.95) = 12.87, *p* < 0.001). Comparing Least-square means (LSM) for treatment effects over time (week 104 vs. week 1) across progressive groups showed lower LSM for the Rapid progressive than the All Progressive group for treatment (lanabecestat 20 mg, lanabecestat 50 mg) compared to placebo (Supplementary Table S5). In particular, we

**A**

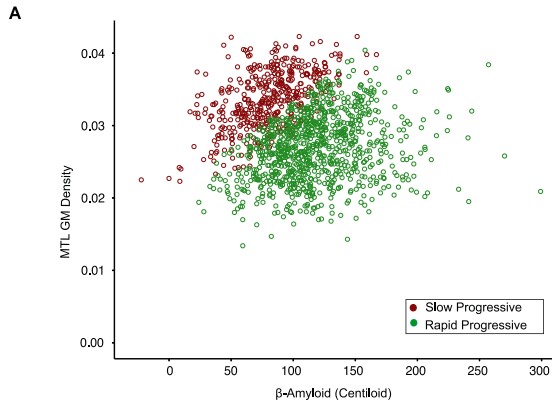

**B**

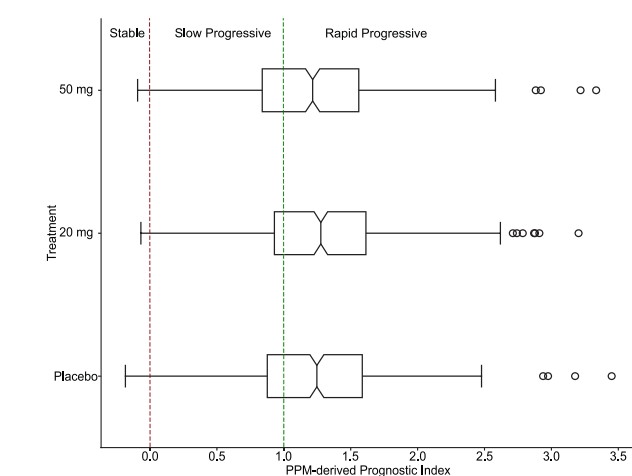

**Fig. 2 | PPM-based stratification of patients in the AMARANTH trial using baseline data. A** Scatter plot of β-Amyloid against MTL GM density for AMARANTH baseline data (week 1) for placebo, lanabecestat 20 mg, lanabecestat 50 mg. Red dots indicate data for Slow progressive; green dots indicate data for Rapid progressive individuals. Individuals with higher β-Amyloid and lower GM density are stratified as Rapid progressive by the PPM model. Individuals with lower β-Amyloid and higher GM density are stratified as Slow progressive by the PPM model. **B** PPM-derived prognostic index for AMARANTH data at baseline (week 1): Box plots showing no significant differences in the PPM-derived prognostic index between treatment groups (placebo, lanabecestat 20 mg, lanabecestat 50 mg) at baseline (sample size: Supplementary Table S2, S3). Notches in the box plots indicate the median, the solid black box represents the 25th to 75th percentile, the black horizontal lines represent the range of the data, and black circles indicate outliers. PPM-derived prognostic index below 0 indicates stable, above 1 indicates rapid progressive, and between 0 and 1 indicates slow progressive individuals. Dashed lines indicate boundaries between stable vs. slow progressive (red) and rapid progressive (green) based on a multinomial logistic regression testing the relationship of the PPM-derived prognostic index to the rate of future tau accumulation. Source data are provided as a Source Data file.

## PPM-guided stratification in the AMARANTH trial shows treatment effects on cognitive outcomes

We tested whether stratifying the AMARANTH dataset based on the PPM-derived index at baseline (week 1) shows treatment effects on cognitive outcomes (CDR-SOB, ADAS-Cog13). In particular, we used a mixed model for repeated measures (MMRM; Supplementary Table S4), to test Treatment (placebo, 20 mg, 50 mg) effects across timepoints (week 1, 52, 104) for each PPM-stratified group (Slow vs. Rapid progressive).

We observed an overall increase in CDR-SOB scores over time (week 104 vs. week 1), suggesting progression in dementia symptoms (Fig. 4A). However, for the Slow progressive group, we observed a significant decrease in CDR-SOB scores for lanabecestat 50 mg compared to placebo at week 104, suggesting slowing down of dementia progression (Fig. 4A). In particular, MMRM analysis showed: a) significant interactions: PPM-stratified group x Treatment x Timepoint (F(4, 2173.3) = 2.62, $p < 0.05$), Treatment × Timepoint (F(4, 2172.8) = 2.20, $p = 0.067$), PPM-stratified group x Treatment (F(2, 1866.4) = 2.51, $p = 0.08$), PPM-stratified group x Timepoint (F(2, 2163.7) = 14.72, $p < 0.001$); b) significant main effects of PPM-stratified group (F(1, 1809.1) = 35.70, $p < 0.001$), Treatment (F(2, 1870.5) = 3.00., $p = 0.05$), Timepoint (F(2, 2190.7) = 24.08, $p < 0.001$). Post-hoc comparisons showed that PPM-stratified group x Timepoint interaction was significant for the lanabecestat 20 mg (F(2, 716.64) = 4.301, $p = 0.014$) and lanabecestat 50 mg (F(2, 709.67) = 15.28, $p < 0.001$) but not the placebo (F(2, 720.26) = 0.96, $p = 0.38$) group.

Further, computing change in CDR-SOB scores over time (week 104 minus week 1) corroborated these results showing significant reduction in CDR-SOB scores (Welch's Two Sample $t$ test, t (84.762) = 2.4475, $p = 0.016$) for lanabecestat 50 mg compared to placebo for the slow progressive group (Fig. 4B). In particular, the slow progressive group showed 33.64 % change in CDR-SOB scores over time compared to higher change in the a) Rapid progressive (80.84%) in the lanabecestat 50 mg group, b) the Slow (77.13%) and Rapid (70.41 %) in the placebo group. That is, the slow progressive group treated with lanabecestat 50 mg showed 46% reduction in cognitive decline compared to the placebo group, suggesting slowing of dementia progression for individuals stratified by the PPM as slow progressive. In contrast, no significant differences in CDR-SOB scores were observed for a) lanabecestat 50 mg compared to placebo in the rapid progressive group t(153.96) = − 0.26, $p = 0.80$, b) lanabecestat 20 mg compared to placebo in the slow (t(66.073) = 0.41, $p = 0.68$) or rapid (t (186.19) = − 0.53, $p = 0.60$) progressive group.

Similar MMRM analyses for the All Progressive group (n = 1354, including individuals in the slow and rapid progressive groups) showed significant main effects of Timepoint (F(2,2195.0) = 23.68, $p < 0.001$) and Treatment (F(2,1883.7) = 3.68, $p = 0.03$), but no significant Treatment x Timepoint interaction (F(4, 2176.9) = 2.00, $p = 0.09$). Further, we did not observe significant changes in CDR-SOB scores over time (week 104 minus week 1) for lanabecestat 20 mg (Welch's Two Sample $t$ test, t (266.93) = − 0.35, $p = 0.72$) nor lanabecestat 50 mg (Welch's Two Sample $t$ test, t (242.09) = 1.16, $p = 0.25$) compared to placebo. These results are consistent with the lack of significant slowing of cognitive decline due to treatment, as previously reported for the AMARANTH trial[21].

Comparing Least-square means (LSM) for treatment effects over time (week 104 vs. week 1) across progressive groups showed lower LSM for the Slow progressive than the Rapid and All Progressive group for treatment compared to placebo (Supplementary Table S5). In particular, we observed the lowest LSM for lanabecestat 50 mg in the Slow Progressive group (LSM = 1.03, SE = 0.25; placebo group LSM = 2.21, SE = 0.24), compared to the Rapid Progressive group (LSM = 2.86, SE = 0.20); placebo group LSM = 2.74, SE = 0.18) and the All Progressive group (LSM = 1.97, SE = 0.16), consistent with the result observed when

observed the lowest LSM for lanabecestat 50 mg for the Rapid Progressive group (LSM = − 25.38, SE = 3.28; placebo group: LSM = − 3.71, SE = 2.90) compared to the Slow Progressive group (LSM = −14.04, SE = 3.10; placebo group: LSM = 3.48, SE = 3.21) and All Progressive group (LSM = − 19.674, SE = 2.43), consistent with the result previously reported for the AMARANTH trial[21] (LSM = − 19.74, SE = 1.97).

Taken together, these results suggest that PPM-derived stratification provides a more sensitive tool for assessing treatment effects, showing stronger β-Amyloid reduction due to lanabecestat treatment in the rapid progressive group than across all progressive individuals or the whole sample considered in the AMARANTH trial.

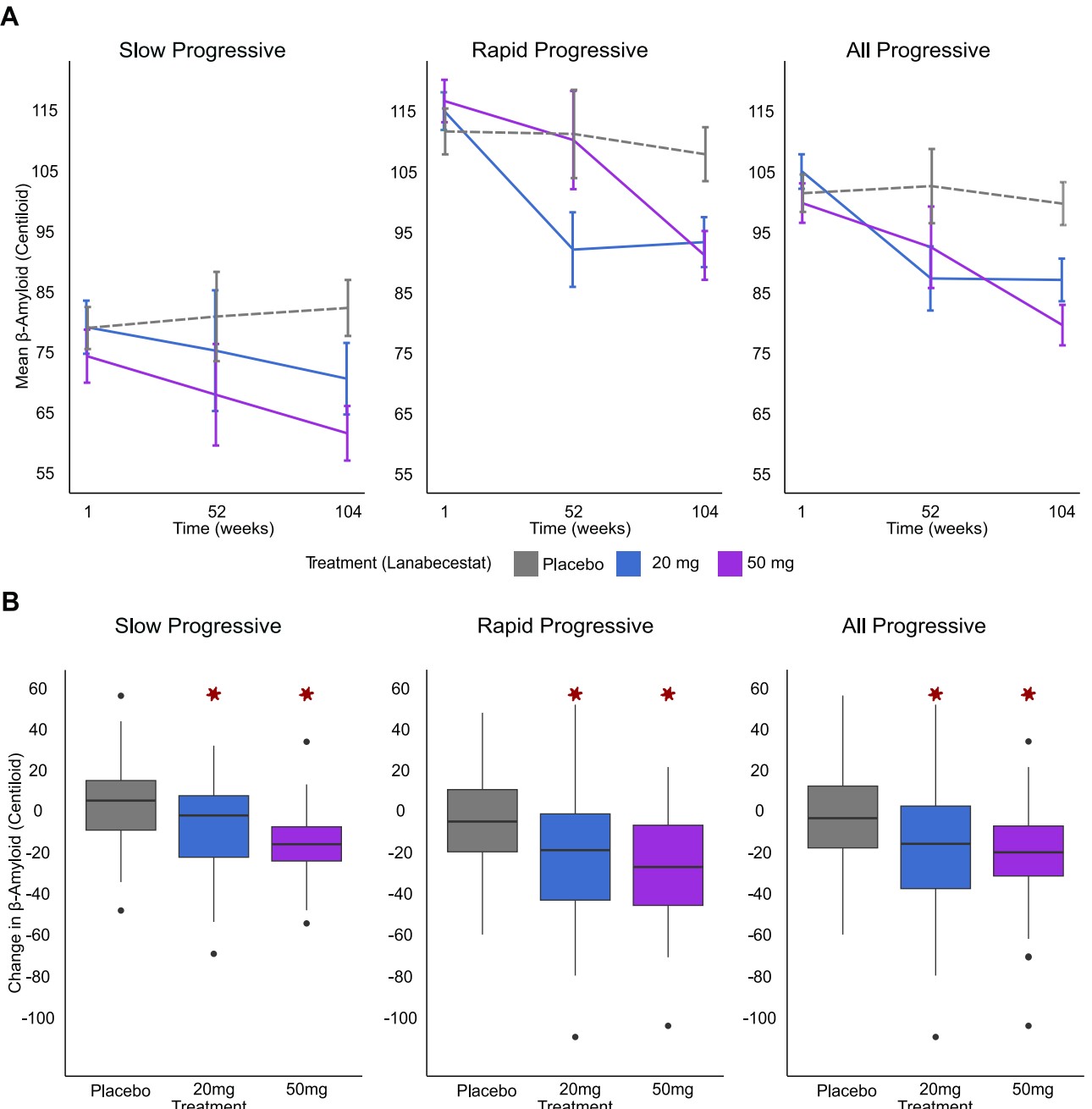

**Fig. 3 | Treatment with lanabecestat in the AMARANTH trial decreases significantly β-Amyloid load for both the slow and rapid progressive individuals.**
**A** Mean β-Amyloid levels over time for Slow, Rapid, and All Progressive individuals (sample size: Supplementary Table S2) in the placebo (gray dashed), lanabecestat 20 mg (blue), and lanabecestat 50 mg (purple). Error bars indicate the standard error of the mean across individuals (SEM). **B**. Box plots of change in β-Amyloid levels (week 104 minus week1) for Slow, Rapid, and All Progressive (sample size: Supplementary Table S2). Black lines in the box plots indicate the median for placebo (gray), lanabecestat 20 mg (blue), and lanabecestat 50 mg (purple), the solid black box represents the 25th to 75th percentile, the black vertical lines represent the range of the data, and black circles indicate outliers. Asterisks indicate significant differences between treatment groups and placebo. Source data are provided as a Source Data file.

the whole clinical trial sample was previously analyzed (LSM = 3.17, SE = 0.18).

Finally, MMRM analyses showed a similar reduction (39%) in ADAS-Cog13 scores for the slow progressive group compared to placebo (Supplementary Fig. S1A). In particular, we observed a significant PPM-stratified group x Timepoint interaction (F(2, 2162.8) = 10.37, $p < 0.001$) but no other significant interactions (Supplementary Table S4). Post-hoc comparisons showed that PPM-stratified group x Timepoint interaction was significant across all groups (lanabecestat 20 mg: F(2, 719.95) = 3.31, $p = 0.03$, lanabecestat 50 mg: F(2,

706.74) = 6.83, $p < 0.001$, placebo: F(2, 720.14) = 4.53, $p = 0.011$), suggesting that the lack of significant treatment effect may be due to differences over time in ADAS-Cog13 in the placebo group. For lanabecestat 50 mg, computing change in ADAS-Cog13 scores over time (week 104 minus week 1) showed lower change (18.62 %) for the Slow progressive group than the Rapid progressive (42.99 %) and for Slow (30.77%) and Rapid (33.62%) in the placebo group; that is, for the Slow progressive group we observed similar treatment effects as for CDR-SOB. However, this reduction in ADAS-Cog13 scores (39% reduction compared to placebo) was not statistically significant (Welch's Two

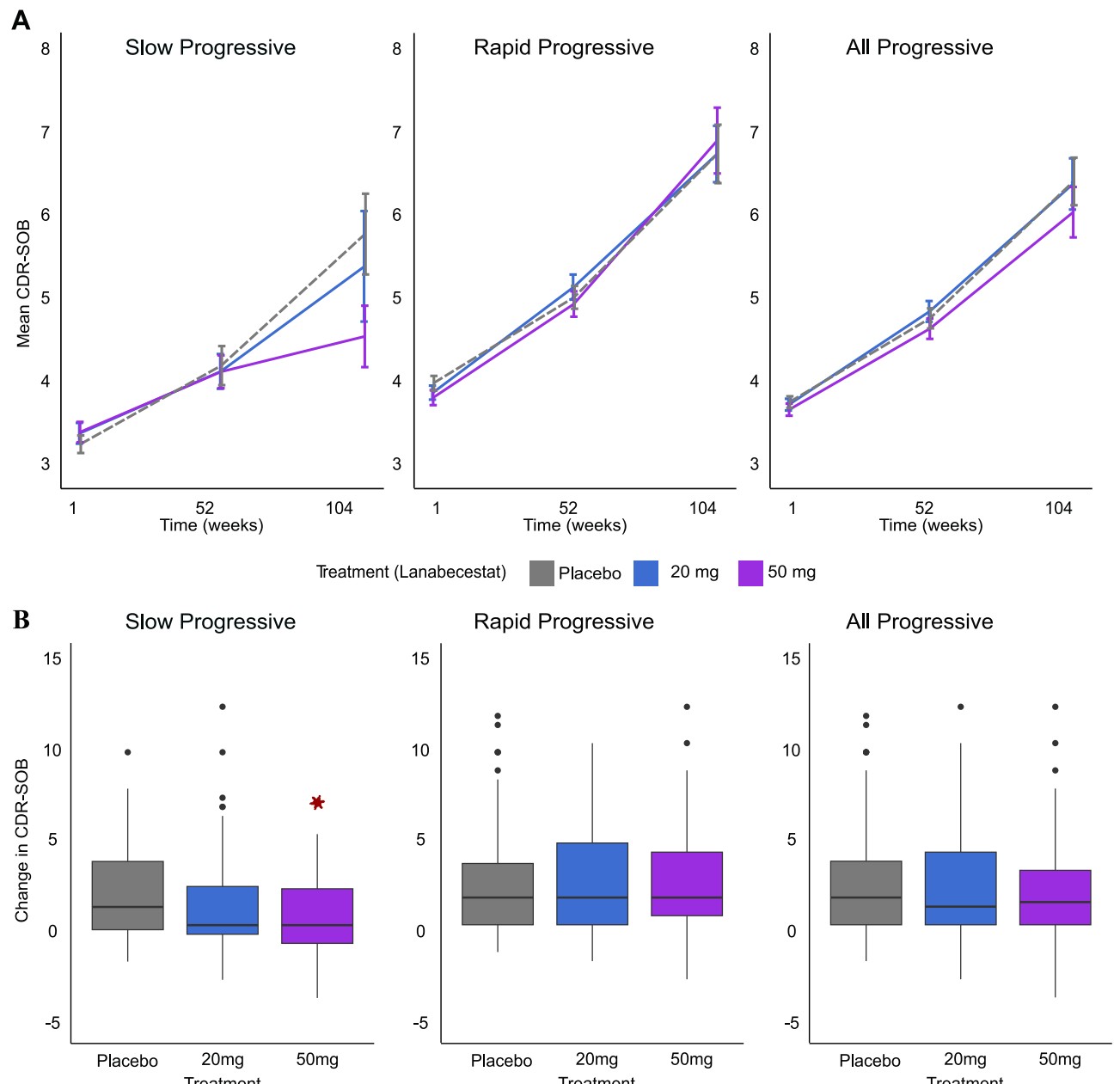

**Fig. 4 | Treatment with lanabecestat in the AMARANTH trial decreases significantly CDR-SOB scores for the slow but not the rapid progressive individuals. A** Mean CDR-SOB over time for Slow, Rapid, and All Progressive individuals (sample size: Supplementary Table S3) in the placebo (gray dashed), lanabecestat 20 mg (blue), and lanabecestat 50 mg (purple). Error bars indicate the standard error of the mean across individuals (SEM). **B** Box plots of change in CDR-SOB (week 104 minus week1) for Slow, Rapid, and All Progressive (sample size: Supplementary Table S3). Black lines in the box plots indicate the median for placebo (gray), lanabecestat 20 mg (blue), and lanabecestat 50 mg (purple), the solid black box represents the 25th to 75th percentile, the black vertical lines represent the range of the data, and black circles indicate outliers. Asterisks indicate significant differences between slow progressive individuals in the 50 mg treatment group vs. placebo. Source data are provided as a Source Data file.

Sample $t$ test, t (78.854) = 0.90, $p$ = 0.37) for lanabecestat 50 mg compared to placebo for the slow progressive group (Supplementary Fig. S1B).

Considering LSM for treatment effects over time (week 104 vs. week 1) showed similar results with the analysis of CDR-SOB. That is, LSM was lower for the Slow progressive than the Rapid progressive, All Progressive group and the whole clinical trial sample for lanabecestat 50 mg compared to placebo (Supplementary Table S5). Taken together, these results suggest that PPM-derived stratification provides a more sensitive tool for assessing treatment effects, providing evidence for slowing of cognitive decline in the slow rather than the rapid

progressive group; that is, lanabecestat 50 mg may slow disease progression at earlier stages of neurodegeneration.

### PPM-guided stratification in the AMARANTH trial changes with treatment

We asked whether treatment has an effect on PPM-guided stratification; that is, whether patients stratified as slow progressive using the PPM-derived prognostic score at baseline (week 1) remain in the slow progressive or transition to the rapid progressive group when stratified based on the PPM-derived prognostic score at week 104. For the placebo, we expected that individuals will transition from the slow to

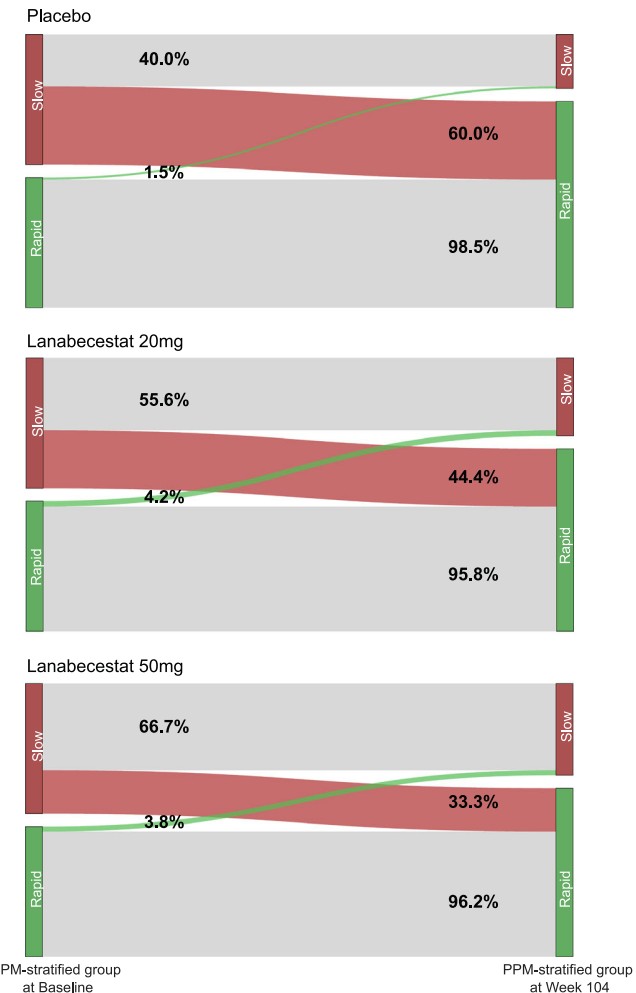

**Fig. 5 | Alluvial plot illustrating changes in PPM-guided stratification due to lanabecestat treatment in the AMARANTH trial.** Percentage of patients in each treatment group (placebo, lanabecestat 20 mg, lanabecestat 50 mg) transitioning between PPM-stratified groups (slow vs. rapid progressive) from baseline to week 104. There is a reduction in the percentage of patients transitioning from the slow progressive group (red) at baseline to the rapid progressive group (green) at week 104 compared to placebo. Conversely, there is an increase in the percentage of patients transitioning from the rapid progressive group at baseline to the slow progressive at week 104 compared to placebo. Source data are provided as a Source Data file.

the rapid progressive group due to neurodegeneration. We reasoned that treatment may slow dementia progression, resulting in a lower number of individuals in the slow group transitioning to rapid progression.

Our results showed that treatment decreased the PPM-derived score compared to placebo, suggesting that treatment slowed cognitive decline (Fig. 5). In particular, a three-way ANOVA on the PPM-derived scores showed a significant Treatment x Timepoint interaction ($F_{(2, 562)} = 3.16$, $p < 0.05$) but not a significant PPM-stratified group x Treatment x Timepoint x ($F_{(2, 562)} = 0.08$, $p = 0.92$). That is, we observed a higher increase in the PPM-derived scores in the placebo group rather than the treatment (lanabecestat 20 mg, lanabecestat 50 mg) groups over time. In particular, for the placebo group, 60% of individuals in the slow progressive group transitioned to rapid progressive, consistent with increased neurodegeneration over time. Treatment with lanabecestat 20 mg decreased this to 44.4% ($\chi^2$ (1) = 7.36, $p < 0.01$) while lanabecestat 50 mg to 33.3% (lanabecestat 50 mg vs. placebo: $\chi^2$ (1) = 7.36, $p < 0.001$). In contrast, individuals in

the placebo group stratified as rapid progressive at baseline (week 1) remained mostly in this group rather than transitioning to the slow progressive group; that is, the percentage of individuals that remained in the rapid progressive group across weeks (week 1, 104) were 98.5% for the placebo, 95.8% for the lanabecestat 20 mg and 96.2% for the lanabecestat 50 mg group. Taken together, these results provide additional evidence that lanabecestat 50 mg may slow dementia progression at earlier stages of neurodegeneration, that is for individuals stratified by the PPM at baseline as slow rather than rapid progressive.

### PPM-guided stratification in the AMARANTH trial decreases sample size necessary for treatment effects on cognitive outcomes

In light of our findings showing that lanabecestat 50 mg may slow dementia progression as measured by a decrease in CDR-SOB scores in the slow progressive group, we asked whether PPM-guided stratification reduces the sample size necessary for future clinical trials. We conducted power calculations to estimate the sample size needed when comparing lanabecestat 50 mg to placebo (i.e., decrease in CDR-SOB change between week 104 and week 1). Figure 6 shows that including slow progressive individuals based on PPM-guided stratification, reduces drastically the sample size necessary.

In particular, for the slow progressive group, we observed a significant moderate effect size (Cohen's $d = 0.51$) for CDR-SOB change in lanabecestat 50 mg vs. placebo ($t(84.76) = 2.45$; $p = 0.016$). We estimated that for this effect size, a sample size of n = 82 per group (lanabecestat 50 mg vs. placebo) would be required for 90% power at alpha 0.05, ($n = 117$ at alpha = 0.01, $n = 164$ at alpha 0.001). In contrast, power calculations for all progressive group (i.e., including both slow and rapid progressive individuals) showed that a sample size of $n = 762$ per group would be required for a small effect size (Cohen's $d = 0.15$); i.e., significant change in lanabecestat 50 mg vs. placebo and 90% power at alpha 0.05 ($n = 1198$ at alpha = 0.01, $n = 1760$ at alpha 0.001). This is consistent with the lack of significant treatment effect on cognitive outcomes that was observed in the AMARANTH trial ($n = 1380$ for both lanabecestat 50 mg and placebo). Redesigning the AMARANTH trial to include only individuals stratified by the PPM at baseline as slow progressive would result to 90.00% reduction in sample size at alpha 0.01 (lanabecestat 50 mg vs. placebo at week 104), compared to the sample size for the all progressive group included in the AMARANTH trial.

### Discussion

Recruiting the right patients at early disease stages is key to efficient clinical trials, maximizing the potential to reveal treatment effects. Patient heterogeneity and lack of sensitive tools for stratification at early stages of dementia pose a major challenge for AD drug discovery. To address this challenge, we built a robust and interpretable clinical-AI tool (PPM) based on a multimodal machine learning approach that supports feature extraction, precise patient classification to clinically stable vs. progressive and trajectory modeling to derive individualized patient prognosis. We have previously shown that the PPM predicts progression to AD at early dementia stages more precisely than standard clinical markers (i.e., gray matter atrophy, cognitive decline, β-amyloid positivity) or clinical diagnosis[18–20]. Here, we demonstrate that the PPM provides a robust tool for patient stratification and inclusion/exclusion in clinical trials with strong potential impact for drug discovery in the following key respects.

First, we demonstrate that the PPM provides a more sensitive tool for patient stratification than standard approaches used for patient selection in clinical trials (e.g., β-amyloid positivity). It is important to know that the PPM was pretrained on research data (ADNI) and tested on the AMARANTH data (independent sample), supporting its utility as a patient stratification tool for clinical trials. Using the PPM to re-stratify the patients included in the AMARANTH trial showed

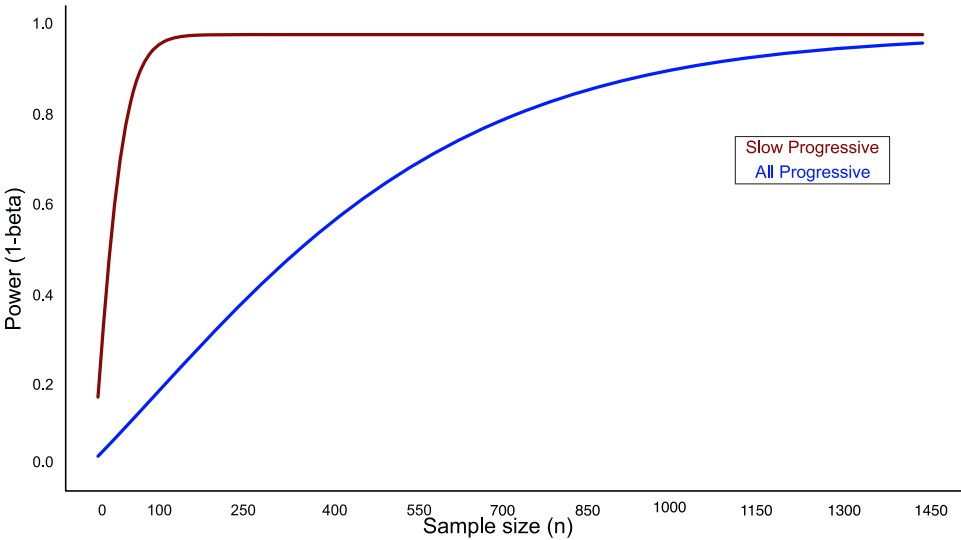

**Fig. 6 | Power calculations show decreased sample size for detecting reduction in cognitive decline with lanabecestat treatment in the AMARANTH trial.** Power (1-beta) of detecting a significant treatment effect (lanabecestat 50 mg vs. placebo) for CDR-SOB change (week 104 minus week 1) at different sample sizes and alpha level of 0.05. Smaller sample size (90% redcution) is needed to detect treatment effects on cognitive outcomes for the Slow (red; effect size: Cohen's $d = 0.51$) compared to the All Progressive (blue; effect size: Cohen's $d = 0.15$) group. Source data are provided as a Source Data file.

treatment effects on outcomes, despite the fact that the trial had been deemed futile. In particular, the PPM predicted whether patients would progress to AD slowly vs. rapidly based on baseline data (i.e., week 1 before treatment). Testing the effect of treatment over 24 months (lanabecestat 20 mg vs. lanabecestat 50 mg) provides evidence that the slow progressive group showed not only a reduction in β-amyloid but, importantly 46% reduction in cognitive decline (i.e., decrease in CDR-SOB) compared to placebo. This is in contrast to the rapid progressive group that showed higher reduction in β-amyloid but no significant difference in cognitive outcomes compared to placebo. We observed a similar slowing of cognitive decline (39%) for ADASCog-13, consistent with previous trials suggesting that ADAS-Cog may be less sensitive in capturing cognitive decline at early dementia stages[25]. Our results are consistent with the findings from—recently FDA and MHRA approved—anti-amyloid antibody target (lecanemab, donanemab) phase 3 trials[9,10]; e.g., lecanemab was shown to slow the rate of cognitive decline by 27%[9]. Although these are independent trials that cannot be directly compared, our results suggest that our AI-guided patient stratification has strong potential to enhance the efficacy of trials and aid the discovery of new treatments.

Second, individuals stratified by the PPM as slow progressive were shown to be at earlier stages of neurodegeneration based on baseline measurements (i.e., higher gray matter density, lower β-amyloid and better cognitive scores) compared to individuals in the rapid progressive group. AD develops gradually, involving a cascade of pathophysiological events beginning with the deposition of β-amyloid that may promote widespread pathological tau protein accumulation, leading to neurodegeneration and cognitive impairment[26]. It is likely that individuals in the rapid progressive group had progressed to later stages of neurocognitive decline, when it was too late for the lanabecestat treatment to be effective in slowing cognitive decline, despite the fact that it resulted in β-amyloid reduction. Interestingly, most patients in the trial were stratified by the PPM as rapid progressive; therefore, when all individuals in the trial were considered, the results were similar to the rapid progressive group; that is, reduction in β-amyloid but no significant changes in cognitive decline were observed. This is consistent with recent work suggesting that the timing of β-amyloid removal during a clinical trial is key for the trial success. If patients are too advanced at the start of the trial, or the underlying disease has progressed to a more advanced stage during the course of

treatment because β-amyloid was not removed early enough, treatments may be less effective[3]. Evidence from previous trials of both successful and unsuccessful anti-β-amyloid therapies suggests that there is little clinical effect of amyloid removal if patients progressed past the mild dementia phase[16,27]. Further, recent clinical trial results (i.e., lecanemab, donanemab)[9,10] showing decline in cognitive symptoms highlight the importance for treatments earlier in disease progression before damage has settled in the brain and when treatments may be more likely to be effective[11,12]. Finally, previous phase II and III trials of BACE inhibitors have shown a robust relationship between dose and degree of β-amyloid reduction, suggesting that a lower dose may be more effective at early stages of the disease[28]. Our results showing slowing of cognitive decline for the higher lanabecestat dose (50 mg vs. 20 mg) for the slow progressive group suggest that higher dose of BACE inhibition may be more effective over a 24-month treatment period, as in the AMARANTH trial.

Our modeling approach has the potential to redefine inclusion/exclusion criteria for clinical trials, reducing patient heterogeneity that is known to hamper statistical power[29]. We have previously shown[20] that our multimodal clinical AI marker (i.e., PPM-derived prognostic index) that captures the multivariate relationships across predictors is more sensitive in early prediction and prognosis of AD than standard clinical markers (β-amyloid, gray matter atrophy, cognitive scores). This is consistent with previous work demonstrating the benefit of integrating multimodal biomarkers for predicting future changes in cognition[23,30–34]. Further, PPM implements a trajectory modeling approach extending beyond binary classifications[22] based on clinical labels (e.g., CN vs AD) that are poorly constrained; that is, individual patients at the class boundary that differ only slightly in their trajectory may be misclassified[35]. In contrast, our PPM-derived clinical AI marker provides a continuous index of future cognitive health from baseline data, reducing misdiagnosis associated with clinical labels and aiding patient stratification based on prognosis (i.e., predicted progression to AD).

Third, the GMLVQ framework with ensemble learning that we adopt in our PPM enables us to develop: a) robust models[36,37] by combining data from multiple disease-relevant modalities, rather than considering single data types, b) interpretable models for early dementia prediction and prognosis, that is key for trusted clinical-AI solutions. In particular, interrogating the model metric tensors allows

us to assess the contribution of different features (i.e., predictors and their interactions) to patient stratification and determine the most predictive data types to include in clinical trials. Further, estimating the distance (based on the trained PPM metric tensors) of an independent test dataset (AMARANTH sample) from the Clinically Stable prototype allows us to predict individual patient trajectories, re-stratify the trial sample and test treatment effects in slow vs. rapid progressive groups. This has the potential to accelerate clinical trials and reduce the costs associated with data collection by (a) focusing on key data types to be collected for patient stratification and treatment outcome assessment, (b) tailoring treatment targets to the right patient groups at different stages of disease progression. Integrating novel markers (e.g., blood biomarkers, digital markers from wearable technologies) for dementia diagnosis into our multimodal modeling approach provides the potential to move to less-invasive and more cost-effective clinical trials.

Finally, we show that using PPM for patient stratification substantially decreases the sample size necessary for identifying significant changes in treatment outcomes. Recruitment and retention of a large number of qualified, diverse volunteers to participate in clinical research studies remain key barriers to the successful completion of AD clinical trials[7,8]. Thus, identifying the right patients at earlier stages for neurodegeneration to include in clinical trials decreases sample heterogeneity and size with strong potential to enhance trial efficiency (faster and cheaper). A possible limitation is that the PPM was trained with biomarker data (β-amyloid, gray matter density) from PET and MRI scans. We have previously shown that PPM reliably predicts cognitive decline when trained with cognitive data alone; yet, adding biomarker data enhances the precision of patient[18–20]. Future work is needed to test whether training the PPM with less-invasive and cost-effective data types (e.g., blood markers, cognitive tests) would provide a reliable stratification tool, reducing further costs and patient burden, and enhancing the efficiency of AD clinical trials.

In sum, there is increasing interest in adopting AI tools for clinical trial optimization and drug discovery[13,38–42]. With trials lasting around 18 months for dementia that typically spans decades, enriching clinical trials with the right patients for specific targets is fundamental to clinical trial outcomes, may account for past failures[43] and inform go / no-go decisions[44]. Clinical AI tools have the potential to play a key role in improving trial design by assisting trial enrichment with the right patients classified prior to enrollment based on AI-guided stratification. Further, including data from underrepresented groups that may be disproportionately affected by dementia is key for tackling the global dementia challenge and developing precision medicine interventions. Our vision is to scale up our predictive prognostic modeling approach to a responsible AI-guided stratification system that will support smarter multi-arm multi-stage trials, accelerating new target discovery for dementia treatment.

## Methods
### PPM training and test samples
We used data from: (1) a research cohort (the Alzheimer's Disease Neuroimaging Initiative, ADNI) for PPM training with within-sample cross-validation ($n = 256$) and independent test ($n = 419$), (2) a Randomized Clinical Trial cohort (AMARANTH, $n = 1354$), as independent test dataset for out-of-sample validation (see Supplementary Material for more information on ADNI and AMARANTH samples, including Patients, Randomization, and Blinding). All data were collected in accordance with ethics approvals at each site and following ethical guidelines (Declaration of Helsinki), including informed consent from participants and approved by the ethics committees at each site.

AMARANTH is a phase 2/3, multicenter, randomized, double-blind, placebo-controlled, global clinical trial of Drug Substance: lanabecestat[21]. Lanabecestat is a brain-permeable inhibitor of human Beta-site amyloid precursor protein-cleaving enzyme 1 (BRACE1/β-

secretase). Patients enrolled in the trial were diagnosed with MCI due to AD or mild AD. The objective of the trial was to test the efficacy of lanabecestat 20 mg lanabecestat and lanabecestat 50 mg (compared to placebo) in slowing AD decline at the end of the double-blind, placebo-controlled period compared to baseline. AD decline was measured by changes in cognition (primary outcome: ADAS-Cog13: 13-item Alzheimer Disease Assessment Scale–cognitive subscale; secondary outcomes: CDR-SB: Clinical Dementia Rating–sum of boxes, MMSE: Mini-Mental State Examination). AMARANTH was terminated early (approximately 16 months prior to planned completion) due to futility analysis. Because of this early termination, we included data from the placebo-controlled periods of the study (weeks: 1, 52, 104; Supplementary Table S2, S3) from patients with: APOE4 at week 1, structural MRI, florbetapir PET (β-amyloid) scans at three time points (week 1, 52, 104), and cognitive measures (CDR-SOB, ADAS-Cog13) at three time points (week 1, 52, 104).

ADNI and AMARANTH datasets differ in patient demographics and data collection tools, allowing us to test PPM interoperability across research cohorts and clinical trial data. In particular, for ADNI, patients were selected with specific criteria related to amnestic MCI and Alzheimer's disease and MRI data were collected across MRI acquisition sites in the US. For AMARANTH, data were collected from patients diagnosed with early AD (i.e., patients with MCI; MCI due to AD) and patients diagnosed with mild dementia of the Alzheimer's type in Australia, Belgium, Canada, the USA, France, Germany, the United Kingdom, Italy, Japan, and Poland. We have previously demonstrated PPM interpretability across diverse research and clinical cohort data collected across sites and countries[20].

### Predictive prognostic modeling
We have developed a trajectory modeling approach based on Generalized Metric Learning Vector Quantization (GMLVQ)[18–20] that leverages multimodal data to make predictions about future cognitive decline at early dementia stages by iteratively adjusting class-specific prototypes and learning class boundaries (Supplementary Material: Predictive Prognostic Model).

Generalized Learning Vector Quantization (GLVQ) is a supervised classification method that iteratively modifies class-specific prototypes to identify boundaries between discrete classes. The GLVQ classifiers are defined by a set of vectors known as class prototypes that represent the classes within the input space. During the training phase, the prototypes are updated iteratively based on the training examples. For each training example, the GLVQ classifier determines the closest prototype for each class. The prototypes are then adjusted so that the prototype representing the same class as the input example (the closest 'correct' prototype) is moved closer to the example, while the closest prototype among the prototypes representing different classes (the closest 'incorrect' prototype) is moved further away. During training, for each class, the GLVQ algorithm aims to minimize the distances between the training examples of the given class and the prototypes that share the same class label, while maximizing the distances to the prototypes of the other classes. This process helps to form class boundaries with large classification margins. Once the training is completed, the GLVQ classifier can be used for classifying test data. Given a previously unseen input vector, the classifier assigns to it the class label of the closest prototype.

GMLVQ is an extension of the GLVQ algorithm that learns the metric to be used in the input space that enhances class separation. The learnt metric is determined through the corresponding metric tensor. GMLVQ incorporates a full metric tensor to provide a robust distance measure (metric) tuned to the classification task. This metric defines a distance that naturally groups together members of the same class while separating the different classes away from each other. Mathematically, it provides specific feature scaling and quantifies pairwise task-conditional dependencies of the input features. Diagonal

elements of the metric tensor identify key predictors, while the off-diagonal terms reveal pairwise feature interactions contributing to the classification task.

Following our previous work[18,19], we trained GMLVQ models to discriminate Clinically Stable (CN individuals who remain stable for 4+ years following baseline; $n = 100$) vs. Clinically Declining (individuals have a baseline diagnosis (at date of FBP scan) of either CN ($n = 17$) or MCI ($n = 139$) but received a diagnosis of dementia in future clinical evaluation (i.e., progressed to dementia ($n = 75$)), or had been diagnosed with dementia in a clinical evaluation prior to baseline (i.e., reverted ($n = 81$); $n = 156$) using ADNI data at baseline: medial temporal lobe gray matter (MTL GM) density[18,19] (see Supplementary Material: MRI analysis: extracting medial temporal gray matter density), β-Amyloid and APOE4. All data were adjusted by regressing out potential confounding covariates (i.e., age, sex, and education). Following our previous work,[18,19] we used the steepest descent method to minimize the cost function through online learning and performed hyperparameter tuning for the model using a nested cross-validation approach[45], considering two hyperparameters. To evaluate the model's performance, we employed 10 iterations of a 10-fold cross-validation[45]. To mitigate any potential biases due to class imbalance in the dataset (Clinically Stable, $n = 100$; Clinically Declining, $n = 156$), we resampled the data to generate balanced classes. For each training fold, we repeatedly ($n = 400$) randomly down-sampled the majority class (i.e., Clinically Declining) to match the size of the minority class (i.e., Clinically Stable). Further, we used ensemble learning[46], combining multiple models ($n = 400$) for robust learning of unbalanced classes. We selected the top 20% ($n = 80$) models based on their training set performance and estimated the class balanced accuracy based on (a) majority vote, i.e., the class label that receives the most votes from the ensemble models is selected as the final prediction[46], (b) the average performance across the selected classifiers[47]. This ensemble learning approach with cross-validation helps mitigate potential individual model biases, resulting in more robust and accurate predictions[36,37].

## PPM-derived prognostic index

Moving beyond binary classifications, we extended the GMLVQ framework to generate continuous predictions for each individual in the test dataset. In particular, we extracted a PPM-derived prognostic index employing a GMLVQ-Scalar Projection[18,19] that extracts distance information (based on the learnt metric tensor) between the sample vector and the learnt class prototypes (representing Clinically stable vs. Declining). GMLVQ-Scalar Projection thus measures the distance as defined by the learnt metric tensor, between an individual and the prototype representing Clinically Stable along the direction separating Clinically Stable vs. Declining (the line connecting stable and progressive class prototypes). We extracted the scalar projection using the average prototypes and metric tensors of the selected top 20% classifiers to capture robust information across the ensemble of trained classifiers. The scalar projection yields a large positive value for Clinically Declining individuals and a zero or negative value for Clinically Stable individuals. That is, the scalar projection index captures information about how far an individual is from the Clinically Stable prototype, serving as an individualized PPM-derived prognostic index. We have previously shown that this index relates significantly to the rate of memory decline[18,20] and future tau accumulation[19], allowing us to estimate how fast an individual progresses from MCI to AD.

Following our previous work [18] we next used multinomial logistic regression to test the relationship of the PPM-derived prognostic index to the rate of future tau accumulation (e.g., future Tau slope in fusiform gyrus that showed significant tau accumulation for Clinically Declining individuals) and determine quartile classes (based on boundaries) that represent different levels of progression. We estimated the probabilities of each quartile class for a range of boundary values and identified the boundaries based on the quartile class with

the highest probability at each value. The lower boundary (at the 20th percentile of the future tau accumulation slope) indicates individuals who are more likely to experience slower progression (slow progressive), while the higher boundary (at the 60th percentile of future tau accumulation) indicates individuals who are more likely to experience faster progression (rapid progressive). This multinomial logistic regression model allows us to stratify individuals based on their PPM-derived prognostic index (i.e., scalar projection score) and future tau accumulation slope.

## Statistics & reproducability

We extracted the PPM-derived prognostic index for each individual in the ADNI test dataset and the AMARANTH dataset. All available data for the PPM input features were used (Supplementary Table S2, S3 for sample size); no statistical method was used to pre-determine the PPM test sample size. We used the Kruskal-Wallis non-parametric test (SPSS) for comparisons of the PPM-derived prognostic index across groups, as the index data were not normally distributed (Shapiro-Wilk test).

To ensure direct comparison to the analysis previously performed on the AMARANTH data[21], we used a mixed model of repeated measures (MMRM), that is typically used in individual-randomized trials with longitudinal continuous outcomes and missing data. We tested the efficacy of lanabecestat 20 mg and lanabecestat 50 mg (compared to placebo) across time points (week 1: baseline, week 52, week 104) on: (a) biomarker outcome: β-amyloid, (b) cognitive outcomes: CDR-SOB, ADAS-Cog13. We conducted this analysis separately for each group (slow progressive, rapid progressive) and for the full sample (All progressive: slow and rapid progressive), including fixed effects for treatment, timepoint, PPM-stratified group. We included the following covariates in the MMRM: covariates for disease status at baseline, age at baseline, APOE4 genotype, baseline outcome measure, AChEI use at baseline, and pooled country. We calculated Least-square means (LSM; emmeans R package) from the MMRM model (mmrm, lme4 R packages) that represent model-adjusted means, including fixed effects for treatment, timepoint and their interactions, while accounting for covariates and unbalanced data. Pairwise comparisons of LSM across treatment groups and timepoints were performed using post-hoc tests, correcting for multiple comparisons. This approach ensures that reported means accurately reflect treatment effects while accounting for within-subject correlations and repeated measurements over time. We repeated these analyses with non-parametric ANCOVA tests (same fixed effects and covariates as for MMRM analyses) to account for deviations from normality (Supplementary Fig. S2, Supplementary Material: Non-parametric Statistical analysis), with similar results as the MMRM analyses.

Further, for each group (slow progressive, rapid progressive) we computed change in outcomes (β-amyloid, cognitive data) at week 104 from baseline (i.e., week 104 minus week 1). We used Welch's $t$ test to test for differences in outcomes across groups (i.e., lanabecestat 20 mg, lanabecestat 50 mg vs. placebo), as it does not assume equal variances between the groups and allows adjusting the degrees of freedom used in the test to accommodate differences in variance. MMRM, post hoc comparisons and Welch's $t$ tests were conducted in R.

Finally, we conducted power analysis, using the 'pwr.t.test' function (pwr package in R), to estimate sample size for change in CDR-SOB (week 104 minus week 1) for lanabecestat 50 mg vs. placebo. A power level of 0.90 (1-β) was chosen to minimize the risk of Type II error, ensuring a high probability of detecting a true treatment effect. We used a standard significance threshold of $\alpha = 0.05$, and more conservative estimates of $\alpha = 0.01$, $\alpha = 0.001$.

## Reporting summary

Further information on research design is available in the Nature Portfolio Reporting Summary linked to this article.

## Data availability

Data may be obtained in accordance with AstraZeneca's data sharing policy described at: https://astrazenecagrouptrials.pharmacm.com/ST/Submission/Disclosure. The raw trial data are protected and are not available due to data privacy laws. Anonymized processed data for studies directly listed on Vivli can be requested through Vivli at www.vivli.org. AstraZeneca Vivli member page is available outlining further details: https://vivli.org/ourmember/astrazeneca/. Data not listed on Vivli could be requested through Vivli at https://vivli.org/members/enquiries-aboutstudies-not-listed-on-the-vivli-platform/. Source data for the figures are provided with this paper and are available at the Cambridge University repository: https://doi.org/10.17863/CAM.117732 Source data are provided in this paper.

## Code availability

Code for figures is available at the Cambridge University repository: https://doi.org/10.17863/CAM.117732.

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

## Acknowledgements

We would like to thank Ali Aghelmaleki for data preparation, Avraam Papadopoulos and Anthony Fernandes for technical support. Data used in the preparation of this article were obtained from the Alzheimer's Disease Neuroimaging Initiative (ADNI) database (adni.loni.usc.edu). As such, the investigators within the ADNI contributed to the design and implementation of ADNI and/or provided data but did not participate in the analysis or writing of this report. A complete listing of ADNI investigators can be found at: http://adni.loni.usc.edu/wp-content/uploads/how_to_apply/ADNI_Acknowledgement_List.pdf. This work was supported by grants to Z.K. from the Royal Society (INF\R2\202107), Alan Turing Institute (TU/B/000095), Wellcome Trust (221633/Z/20/Z). For the purpose of open access, the authors have applied for a CC BY public copyright license to any Author Accepted Manuscript version arising from this submission. Data collection and sharing for the Alzheimer's Disease Neuroimaging Initiative (ADNI) is funded by the National Institute on Aging (National Institutes of Health Grant U19 AG024904). The grantee organization is the Northern California Institute for Research and Education. In the past, ADNI has also received funding from the National Institute of Biomedical Imaging and Bioengineering, the Canadian Institutes of Health Research, and private sector contributions through the Foundation for the National Institutes of Health (FNIH) including generous contributions from the following: AbbVie, Alzheimer's Association; Alzheimer's Drug Discovery Foundation; Araclon Biotech; BioClinica, Inc.; Biogen; Bristol-Myers Squibb Company; CereSpir, Inc.; Cogstate; Eisai Inc.; Elan Pharmaceuticals, Inc.; Eli Lilly and Company; EuroImmun; F. Hoffmann-La Roche Ltd and its affiliated company Genentech, Inc.; Fujirebio; GE Healthcare; IXICO Ltd.; Janssen Alzheimer Immunotherapy Research & Development, LLC.; Johnson & Johnson Pharmaceutical Research &Development LLC.; Lumosity; Lundbeck; Merck & Co., Inc.; Meso Scale Diagnostics, LLC.; NeuroRx Research; Neurotrack Technologies; Novartis Pharmaceuticals Corporation; Pfizer Inc.; Piramal Imaging; Servier; Takeda Pharmaceutical Company; and Transition Therapeutics.

## Author contributions

Conceptualization: D.V., G.M., Z.K., P.T., C.S., A.L., and K.T. Methodology: D.V., G.M., Z.K., P.T., C.S., A.L., and K.T. Investigation: D.V., G.M., Z.K., P.T., C.S., A.L., and K.T. Visualization: D.V. and G.M. Funding acquisition: Z.K. Project administration: Z.K., A.L., and K.T. Supervision: Z.K., P.T., A.L., and K.T. Writing – original draft: D.V., G.M., Z.K., P.T., C.S., A.L., and K.T. Writing – review & editing: D.V., G.M., Z.K., P.T., C.S., A.L., and K.T.

## Competing interests

G.M., C.S., A.L., and K.T. are employees or former employees of AstraZeneca and may own shares. The remaining authors declare no competing interests.
