## [Transparent Peer Review file · Nature Communications]

AI-guided patient stratification improves outcomes and efficiency in the AMARANTH Alzheimer's Disease clinical trial

Corresponding Author: Professor Zoe kourtzi

Version 0:

Reviewer comments:

Reviewer #1

(Remarks to the Author)

This work proposes AI based AD patients' stratification.

Firstly, writing style and presentation need improvements throughout the manuscript.

Results are not very impressive, mainly the difference between sensitivity and specificity. Likewise, existing approaches has been showing much better performance and up-to 97% accuracy.

Figures need axes label and units for better readability

Overall, technical contributions are limited to be published in Nature COMM. The author did utilize well established models to classify/predict the disease progression.

The outcomes and methodology are not reflecting the claim of interpretable AI model

Reviewer #2

(Remarks to the Author)

Review of the Manuscript

This manuscript presents a highly significant and original contribution, to Alzheimer's research and to biomedical research and clinical trials more broadly. The study demonstrates that a machine learning model, trained entirely on one curated dataset (ADNI), can be applied to a separate clinical trial dataset (AMARANTH)—without retraining—going well beyond reliable and meaningful insights: rescuing a clinical trial. This finding is crucial, as it brings evidence that machine learning and AI tools can actually provide robust patient stratification models. Such models can significantly boost performance in clinical trials and bring to the population useful treatments that otherwise would have been discarded. This can be achieved, for instance, by identifying subpopulations that respond to treatment, which conventional analyses may overlook.

Ultimately, this work provides a generalisable framework for precision medicine. The ability of advanced stratification models to refine clinical trial analyses can improve drug development and regulatory decision-making, ensuring that highly specialised treatments reach the right patient groups.

Importantly, the way the model works seems compatible with principles of interpretability and current clinic trials as it seems able to provide guidance from stratification prior to the trial; without further interference to the trial itself. The way the trial was re-analysed shows that if the trial would have yielded results is the groups have been appropriately separated at the very beginning.

This manuscript's novelty lies in the first rigorous demonstration of its robustness beyond the training dataset and applicability in drug trials: it was trained on a research dataset (ADNI) and successfully applied to an independent real-world clinical trial dataset (AMARANTH) without retraining. This cross-dataset generalisability is a strong validation of the model's reliability and translational potential.

Areas for Improvement

To improve clarity and accessibility for both machine learning researchers and clinical practitioners, the following

refinements are recommended:

Major

The most critical element and its interpretation for the clinical community—that the model was trained on ADNI and applied to AMARANTH without retraining—is not sufficiently emphasised in the manuscript. This point should be made clear earlier in the text, as it represents a key contribution of the study.

Minor

1. Clarifications in Figures and Terminology

- Figure 1:
 - Panel A serves as a consistency check for the model, as it aligns with established domain knowledge. This should be explicitly discussed to reinforce confidence in the model's reliability.
 - Panel B would benefit from clearer visual indicators, such as dashed vertical lines at indices 0 and 1, marking key thresholds. A brief explanation of how these thresholds were selected would improve interpretability.
 - The selection of thresholds 0 and 1 should be better explained. While it is clear that they result from normalization, further discussion of how this normalization was performed and why these specific values were chosen would be helpful.
- Figure 2:
 - Panel A: If appropriate, further highlighting the overlapping region between slow-progressing and rapid-progressing patient groups would aid in understanding the separation challenge and the role of additional model components.
 - To improve consistency, drawing the same horizontal lines as in Figure 1, Panel B would help visually link the analysis across datasets, showing the connection between how information from the model relates to both the original and new datasets.
- Figure 3:
 - The figure currently extends beyond the page margins, causing legends to be cut off. Adjusting the layout to reposition the legend would resolve this issue.
- Figure 4 and all figures in general:
 - Each figure should include a clear key message to help guide the reader's interpretation.

2. Clarifications in Data and Analysis

- AMARANTH Clinical Trial Data:
 - The definition of slow progression in AMARANTH patients should be more explicitly highlighted in the main text.
- Threshold Selection and Normalization:
 - The manuscript should elaborate on how the thresholds at 0 and 1 were selected, particularly how normalization was performed and its implications for interpreting model predictions.
- Ground Truth in Clinical Trials:
 - Since the true treatment effect in clinical trials is inherently unknown, further discussion of what evidence supports the model's consistency across datasets would strengthen the manuscript.

3. Explanation of Statistical Models

- The mixed model for repeated measures (MMRM) is used for treatment effect estimation. A brief explanation of why this model was chosen would be beneficial.
- Least Squares Means (LSMeans) are referenced but would benefit from a clearer explanation of how they were calculated.
- A more detailed discussion of the G*Power analysis would be helpful.

Final Recommendation

This manuscript presents a significant demonstration of how machine learning can enhance patient stratification in clinical trials, improving the detection of treatment effects that conventional statistical models might overlook. While the findings are highly relevant for Alzheimer's disease, their implications extend broadly to precision medicine, clinical trial design, and regulatory science. The suggested refinements focus on clarifying key methodological choices and improving accessibility to a wider audience.

I recommend acceptance after addressing these comments.

Reviewer #3

(Remarks to the Author)

Vaghari et al. present an intriguing manuscript wherein they have built a clinical AI-tool (predictive prognostic model, PPM) that was trained on multimodal data to predict a person's progression to AD in early stages of dementia. They trained the PPM using baseline data from ADNI to distinguish individuals that are clinically stable vs those that are clinically declining. To do so they used measures of A β (imaging), APOE4 and medial temporal lobe GM density. Next, they used this approach in an independent ADNI sample and showed that the PPM-derived prognostic index showed subgroups that were more likely to remain stable vs progress rapidly vs progress slowly. Following this, the authors moved to use the ADNI-trained PPM to stratify patients from the AMARANTH study (of BACE1 inhibitor, lanabecestat) into 2 groups: slow vs rapid progressors. They then analyzed the treatment effects on biomarker (A β) and clinical outcomes (CDR-SOB and ADAS-Cog13) with a mixed model for repeated measure for each PPM-stratified group. PPM-derived stratification provided a more

sensitive tool for assessing treatment effects. It appears to show evidence for a slowdown of cognitive decline in the lanabecestat-treated group of slow-progressors, suggesting that the higher drug dose may slow down AD progression at earlier stages. The authors also showed that using PPM-guided stratification for clinical study design could significantly decrease sample size. In the AMARANTH trial, if it had been designed with only slow progressors (who showed the biggest cognitive benefit according to the authors' results), it could have cut down the sample size by 90%.

As there is a lot of heterogeneity in humans, it is critically important to stratify them properly into clinical trials to ensure we do not miss potential benefits of drugs that would otherwise be cast aside due to no benefit in the overall group. Thus, an approach like this could significantly help the field.

There are some changes to the manuscript that are necessary to help make it clearer to the reader:

1. Supplemental Figures/Tables are not in the order they are referenced to throughout the manuscript.
2. Figure 3A: legend is cut off.
3. Please include tables summarizing the statistical treatment effects/significant interactions for A β , CDR-SOB and ADAS-Cog13 (pages 10, 11, 13 & 14).
4. Please indicate the significant findings in figures 3, 4 and S1 with asterisks.
5. In the last paragraph on pg 14, mention the p-value for the ADAS-Cog13 and describe the MMRM interaction results as was done for CDR-SOB (or provide this in supplement as well as a table summarizing this in supplement).
6. On p 16, the 2nd paragraph that describes the transitioning of individuals from one group to the other: please provide a graphical representation of this as well as a table that can be in supplement.
7. Fig 5: the description under the graph says that the 'all progressive' group is 'red' but the legend and the line on the graph for 'all progressive' is green.

Reviewer #4

(Remarks to the Author)

Version 1:

Reviewer comments:

Reviewer #1

(Remarks to the Author)

REJECT: Manuscript does not presents novel contributions to be published in NCOMMs

Reviewer #2

(Remarks to the Author)

I am satisfied with the updated manuscript.

Reviewer #3

(Remarks to the Author)

The manuscript has been significantly improved in this revision. Two minor comments remain:

1. In the new Table S4, please denote the p-values that reach statistical significance with bold text for clarity.
2. On the first line of p. 14, the text reads: "Similar analyses in the All Progressive group (n= 1354, including individuals in the slow and rapid progressive groups) showed significant increase rather than decrease in CDR-SOB scores compared to the placebo group."
 - a. This statement refers to the right-most panel of Figure 4A & B. The "All Progressive" panels appear to show a decrease (rather than an increase) in CDR-SOB scores in the 50 mg group compared to placebo.
 - b. The text mentions this is a significant change, so please add the asterisk to the corresponding panel.

Reviewer #4

(Remarks to the Author)

Response to reviewers

Reviewer #1 (Remarks to the Author):

This work proposes AI based AD patients' stratification.

1. Firstly, writing style and presentation need improvements throughout the manuscript.

We have revised the text to improve clarity and readability.

2. Results are not very impressive, mainly the difference between sensitivity and specificity. Likewise, existing approaches has been showing much better performance and up-to 97% accuracy.

Our model is trained to identify whether an individual will remain stable vs. progress (slowly vs. rapidly) to AD within 4+ years (i.e. clinically stable vs. progressive). This is a finer classification task compared to previous work focusing on patients (AD) vs. healthy (cognitive normal) classifications. Signals for AD vs CN classifications are more discriminable and may result in higher model performance. Thus, achieving higher than 90% accuracy (91.1%) and precision for our finer classification task (clinically stable vs. progressive) following cross-validation provides evidence for the robustness of our modeling approach. We note the difference between sensitivity (87.5%) and specificity (94.2%) that is likely due to weaker signals for the clinically progressive compared to the clinically stable class. However, both rates are above 85% resulting in high precision (93.8%) in support of model reliability.

We have now clarified this point further in the revised text. In particular, the text writes:

‘Note that discriminating Clinically Stable vs. declining individuals comprises a finer classification task compared to previous work focusing on patient (AD) vs. healthy (cognitive normal) classifications^{35,36}, where signals are more discriminable and may result in higher model performance. The difference between sensitivity and specificity is likely due to weaker signals for the clinically progressive compared to the stable class. Thus achieving higher than 90% accuracy and precision provides evidence for PPM robustness.’

3. Figures need axes label and units for better readability

We have revised all figures to be more readable

4. Overall, technical contributions are limited to be published in Nature COMM. The author did utilize well established models to classify/predict the disease progression.

We would like to take the opportunity to clarify the novelty of our approach. We have built a model architecture (PPM) that supports multimodal feature extraction, precise classification of patients to clinically stable vs. progressive and trajectory modelling to derive individual patient prognosis. We have previously validated our model predictions against multiple independent data samples and longitudinal real-world patient data.

The novelty of the work presented in this paper lies in demonstrating that our modelling approach can be successfully applied for patient stratification in clinical trials enhancing their efficiency and outcomes. It is important to note that our model was trained on research datasets (ADNI) and tested on completely independent and truly out-of-sample clinical trial data (AMARANTH), demonstrating the strong potential of our modeling approach for translation in clinical settings. In particular, we use the AMARANTH data as an independent test sample, stratify the patients at baseline into slow vs rapid progressive and demonstrate that patients in the slow progressive group show significant reduction in cognitive decline. This result provides evidence that using our AI-guided tool to re-stratify patients rescues a clinical trial that was deemed futile; that is, if the right patients were included in the trial (i.e. slow progressive patients at earlier neurodegeneration stages) the trial would have met the primary outcome of

reduction in cognitive decline. We believe that our modelling approach and findings make a novel and important contribution with strong translational impact for clinical trials and drug discovery.

We have now clarified this point further in the revised text. In particular, the Discussion section writes:

‘Patient heterogeneity and lack of sensitive tools for stratification at early stages of dementia pose a major challenge for AD drug discovery. To address this challenge, we built a robust and interpretable clinical-AI tool (PPM) based on a multimodal machine learning approach that supports feature extraction, precise patient classification to clinically stable vs. progressive and trajectory modelling to derive individualized patient prognosis. We have previously shown that PPM predicts progression to AD at early dementia stages more precisely than standard clinical markers (i.e. grey matter atrophy, cognitive decline, β -amyloid positivity) or clinical diagnosis¹⁸⁻²⁰. Here, we demonstrate that PPM provides a robust tool for patient stratification and inclusion/exclusion in clinical trials with strong potential impact for drug discovery in the following key respects.

First, we demonstrate that PPM provides a more sensitive tool for patient stratification than standard approaches used for patient selection in clinical trials (e.g. β -amyloid positivity). It is important to know that the PPM was pretrained on research data (ADNI) and tested on the AMARANTH data (independent sample), supporting its utility as patient stratification tool for clinical trials. Using the PPM to re-stratify the patients included in the AMARANTH trial showed treatment effects on outcomes, despite the fact that the trial had been deemed futile.’

5. The outcomes and methodology are not reflecting the claim of interpretable AI model

Our model architecture is transparent and highly interpretable. First, interrogating the metric tensors allows us to understand the contribution of each feature to the model’s prediction, as well as feature interactions (pairwise: off-diagonal terms; higher-order: eigenbasis). Second, the prototypes (one per class: clinically stable, progressive) represent the most discriminative classes and allow us to predict an individual’s trajectory. That is, using the scalar projection we estimate the distance (based on the learnt metric tensor) of a new test dataset from the clinically stable prototype (projected to the significant eigenbasis of the metric tensor) and determine a prognostic index for each individual, allowing individualized prognosis beyond binary clinical labels. Based on this scalar projection index, we then re-stratify individuals to stable, slow vs. rapid progressive groups, allowing us to test treatment effects across groups.

We now discuss this point in more detail in the revised manuscript. In particular, the Results section writes:

‘Further, the PPM architecture is transparent and interpretable. First, interrogating the metric tensors allows us to understand the contribution of each feature to the model’s prediction. In particular, the metric tensors indicate that β -amyloid burden (weight: 0.51) is the most discriminative feature compared to MTL GM density (weight: 0.34) and APOE4 (weight: 0.15). This is consistent with the role of β -amyloid and MTL atrophy as markers of Alzheimer’s pathology, consistent with the NIA-AA 2018 diagnostic framework of AD²³.

Second, interrogating the off-diagonal terms of the metric tensor allows us to understand the feature interactions that contribute to the model’s prediction. In particular, we observed a positive interaction between baseline β -amyloid burden and APOE 4, while a negative interaction between baseline β -amyloid burden and MTL GM density, consistent with the role of β -amyloid and APOE 4 as risk factors for progression to AD and neurodegeneration, as indicated by MTL atrophy (i.e. decrease in MTL GM density).

Third, the PPM prototypes (one per class: clinically stable, progressive) indicate the most discriminative class representatives and allow us to predict an individual's trajectory. That is, using the GMLVQ-Scalar Projection method, we estimate the distance (based on the learnt metric tensor) of a new test dataset from the Clinically Stable prototype and determine the PPM-derived prognostic index for each individual, allowing individualized prognosis beyond binary clinical labels (Figure 1B; Supplementary Material: GMLVQ-Scalar Projection).'

Further, the Discussion section writes:

'Third, the GMLVQ framework with ensemble learning that we adopt in our PPM enables us to develop: a) robust models^{37,38} by combining data from multiple disease-relevant modalities, rather than considering single data types, b) interpretable models for early dementia prediction and prognosis, that is key for trusted clinical-AI solutions. In particular, interrogating the model metric tensors and prototypes allows us to assess the contribution of different features (i.e. predictors and their interactions) to patient stratification and determine the most predictive data types to include in clinical trials. Further, estimating the distance (based on the PPM metric tensors) of an independent test dataset (AMARANTH sample) from the Clinically Stable prototype allows us to predict individual patient trajectories, re-stratify the trial sample and test treatment effects in slow vs. rapid progressive groups. This has potential to accelerate clinical trials and reduce the costs associated with data collection by a) focusing on key data types to be collected for patient stratification and treatment outcome assessment, b) tailoring treatment targets to the right patient groups at different stages of disease progression.'

Reviewer #2 (Remarks to the Author):

Review of the Manuscript

This manuscript presents a highly significant and original contribution, to Alzheimer's research and to biomedical research and clinical trials more broadly. The study demonstrates that a machine learning model, trained entirely on one curated dataset (ADNI), can be applied to a separate clinical trial dataset (AMARANTH)—without retraining—going well beyond reliable and meaningful insights: rescuing a clinical trial. This finding is crucial, as it brings evidence that machine learning and AI tools can actually provide robust patient stratification models. Such models can significantly boost performance in clinical trials and bring to the population useful treatments that otherwise would have been discarded. This can be achieved, for instance, by identifying subpopulations that respond to treatment, which conventional analyses may overlook.

Ultimately, this work provides a generalisable framework for precision medicine. The ability of advanced stratification models to refine clinical trial analyses can improve drug development and regulatory decision-making, ensuring that highly specialised treatments reach the right patient groups. Importantly, the way the model works seems compatible with principles of interpretability and current clinic trials as it seems able to provide guidance from stratification prior to the trial; without further interference to the trial itself. The way the trial was re-analysed shows that if the trial would have yielded results in the groups have been appropriately separated at the very beginning.

This manuscript's novelty lies in the first rigorous demonstration of its robustness beyond the training dataset and applicability in drug trials: it was trained on a research dataset (ADNI) and successfully applied to an independent real-world clinical trial dataset (AMARANTH) without retraining. This cross-dataset generalisability is a strong validation of the model's reliability and translational potential.

Thank you for your positive comments and acknowledging the advantages, importance and novelty of our work.

Areas for Improvement

To improve clarity and accessibility for both machine learning researchers and clinical practitioners, the following refinements are recommended:

Major: The most critical element and its interpretation for the clinical community—that the model was trained on ADNI and applied to AMARANTH without retraining—is not sufficiently emphasised in the manuscript. This point should be made clear earlier in the text, as it represents a key contribution of the study.

Thank you for highlighting this advantage of our approach. We have now highlighted this point throughout the revised manuscript. For example, the text writes:

'It is important to know that the PPM was pretrained on research data (ADNI) and tested on the AMARANTH data (independent sample), supporting its utility as patient stratification tool for clinical trials.'

1. Minor: Clarifications in Figures and Terminology

Figure 1:

• *Panel A serves as a consistency check for the model, as it aligns with established domain knowledge. This should be explicitly discussed to reinforce confidence in the model's reliability.*

Thank you for this suggestion. We have revised the text to emphasise this point.

In particular the text writes: 'This is consistent with the role of β -amyloid and MTL atrophy as markers of Alzheimer's pathology, consistent with the NIA-AA 2018 diagnostic framework of AD²³.'

- *Panel B would benefit from clearer visual indicators, such as dashed vertical lines at indices 0 and 1, marking key thresholds. A brief explanation of how these thresholds were selected would improve interpretability.*

Thank you for this suggestion. We have revised Figure 1 to include vertical dashed lines.

- *The selection of thresholds 0 and 1 should be better explained. While it is clear that they result from normalization, further discussion of how this normalization was performed and why these specific values were chosen would be helpful.*

Thank you for this suggestion. We have revised the text to clarify this better. In particular, the text writes:

‘We then used multinomial logistic regression to capture the relationship of the PPM-derived prognostic index to the rate of future tau accumulation and determine boundaries for quartile classes that differ in likelihood of disease progression. We scaled the boundaries so that PPM-derived prognostic index indicates individuals who are more likely to: 1) remain stable (PPM index values below 0 fall at the 20th percentile of the future tau accumulation slope), 2) experience rapid progression (PPM index values higher than 1 fall at the 60th percentile of future tau accumulation), 3) experience slower progression (PPM index between 0 and 1)^{18,19.}’

Figure 2:

- *Panel A: If appropriate, further highlighting the overlapping region between slow-progressing and rapid-progressing patient groups would aid in understanding the separation challenge and the role of additional model components.*

Thank you for this suggestion. We have revised Figure 2A to demonstrate visually the overlap between slow and rapid progressive

- *To improve consistency, drawing the same horizontal lines as in Figure 1, Panel B would help visually link the analysis across datasets, showing the connection between how information from the model relates to both the original and new datasets.*

Thank you for this suggestion. We have included dashed lines to indicate the boundaries between stable vs. slow progressive and rapid progressive as in Figure 1B.

Figure 3: *The figure currently extends beyond the page margins, causing legends to be cut off. Adjusting the layout to reposition the legend would resolve this issue.*

Thank you for spotting this. We have now revised this figure ensuring the legend appears within the figure space.

Figure 4 and all figures in general: *Each figure should include a clear key message to help guide the reader’s interpretation.*

Thank you for this suggestion. We have revised the figure captions accordingly.

2. Clarifications in Data and Analysis

- AMARANTH Clinical Trial Data: *The definition of slow progression in AMARANTH patients should be more explicitly highlighted in the main text.*

Thank you for this suggestion. We have revised the text to explain this more clearly.

In particular, the revised text writes: “In particular, using the GMLVQ-Scalar Projection method, we estimated the distance of the AMARANTH dataset at baseline (week 1) from the Clinically Stable prototype, determined the PPM-derived prognostic index for each individual in the AMARANTH sample and stratified individuals as slow vs. rapid progressive based on baseline (week 1) data (Figure 2A; the sample size for stable was small (n=5) and these data were excluded from further analysis).”

• *Threshold Selection and Normalization:* The manuscript should elaborate on how the thresholds at 0 and 1 were selected, particularly how normalization was performed and its implications for interpreting model predictions.

Thank you for this suggestion. We have revised the text to clarify this better. In particular the text writes:

‘We then used multinomial logistic regression to capture the relationship of the PPM-derived prognostic index to the rate of future tau accumulation and determine boundaries for quartile classes that differ in likelihood of disease progression. We scaled the boundaries so that PPM-derived prognostic index indicates individuals who are more likely to: 1) remain stable (PPM index values below 0 fall at the 20th percentile of the future tau accumulation slope), 2) experience rapid progression (PPM index values higher than 1 fall at the 60th percentile of future tau accumulation), 3) experience slower progression (PPM index between 0 and 1)^{18,19}.’

• *Ground Truth in Clinical Trials:* Since the true treatment effect in clinical trials is inherently unknown, further discussion of what evidence supports the model’s consistency across datasets would strengthen the manuscript.

Thank you for this suggestion. We have revised the text to clarify this better.

For the independent ADNI test sample, where patients varied in their diagnosis (CN, MCI, AD), we used diagnosis as validation outcome. In particular, the text writes:

‘Our results (Figure 1B) showed that the PPM-derived prognostic index was significantly different across groups (Kruskal-Wallis H test $\chi(2) = 121.46$, $p < 0.001$) with significantly higher index (Bonferroni corrected) for AD vs. MCI and CN ($p < 0.001$), MCI vs. CN ($p < 0.001$). This validation against clinical outcomes (i.e. diagnosis) provides evidence that the PPM-derived prognostic score is clinically relevant.’

For the AMARANTH sample, as all patients were diagnosed as MCI, we used cognitive decline as validation outcome. In particular, the text writes:

‘Interestingly, individuals in the slow progressive group showed lower β -amyloid burden ($t(330.22) = 11.833$, $p < 0.001$), higher MTL GM density ($t(326.33) = -17.351$, $p < 0.001$) and better performance in cognitive tests (i.e. lower CDR-SOB ($t(911.03) = 6.38$, $p < 0.001$) and ADAS-Cog13 ($t(806.77) = 5.23$, $p < 0.001$)) compared to the rapid progressive group at baseline. These results suggest that individuals stratified as slow progressive by the PPM were at earlier stages of neurodegeneration compared to individuals stratified as rapid progressive, corroborating the link of the PPM-derived prognostic score to cognitive decline and disease progression’

3. Explanation of Statistical Models

• *The mixed model for repeated measures (MMRM) is used for treatment effect estimation. A brief explanation of why this model was chosen would be beneficial.*

Thank you for this suggestion. We have revised the text to clarify this better. In particular the text writes: ‘To ensure direct comparison to the analysis previously performed on the AMARANTH data²¹, we used a mixed model of repeated measures (MMRM), that is typically used in individual-randomized trials with longitudinal continuous outcomes and missing data.’

• *Least Squares Means (LSMeans) are referenced but would benefit from a clearer explanation of how they were calculated.*

Thank you for this suggestion. We have revised the text to clarify this better. In particular, the text writes:

‘We calculated Least-square means (LSM; emmeans R package) from the MMRM model (mmrm, lme4 R packages) that represent model-adjusted means, including fixed effects for treatment, timepoint and their interactions, while accounting for covariates and unbalanced data. Pairwise comparisons of LSM across treatment groups and timepoints were performed using post-hoc tests, correcting for multiple comparisons. This approach ensures that reported means accurately reflect treatment effects while accounting for within-subject correlations and repeated measurements over time.’

• *A more detailed discussion of the G*Power analysis would be helpful.*

Thank you for this suggestion. We have revised the text to clarify this better. In particular the text writes:

‘Finally, we conducted power analysis, using the `pwr.t.test` function (pwr package in R), to estimate sample size for change in CDR-SOB (week 104 minus week 1) for lanabecestat 50mg vs. placebo. A power level of 0.90 ($1-\beta$) was chosen to minimize the risk of Type II error, ensuring a high probability of detecting a true treatment effect. We used a standard significance threshold of $\alpha = 0.05$, and more conservative estimates of $\alpha = 0.01$, $\alpha = 0.001$.’

Final Recommendation

This manuscript presents a significant demonstration of how machine learning can enhance patient stratification in clinical trials, improving the detection of treatment effects that conventional statistical models might overlook. While the findings are highly relevant for Alzheimer’s disease, their implications extend broadly to precision medicine, clinical trial design, and regulatory science. The suggested refinements focus on clarifying key methodological choices and improving accessibility to a wider audience.

I recommend acceptance after addressing these comments.

Thank you for the positive and constructive comments, we feel they have improved the clarity and quality of our manuscript.

Reviewer #3 (Remarks to the Author):

Vaghari et al. present an intriguing manuscript wherein they have built a clinical AI-tool (predictive prognostic model, PPM) that was trained on multimodal data to predict a person's progression to AD in early stages of dementia. They trained the PPM using baseline data from ADNI to distinguish individuals that are clinically stable vs those that are clinically declining. To do so they used measures of $A\beta$ (imaging), APOE4 and medial temporal lobe GM density. Next, they used this approach in an independent ADNI sample and showed that the PPM-derived prognostic index showed subgroups that were more likely to remain stable vs progress rapidly vs progress slowly. Following this, the authors moved to use the ADNI-trained PPM to stratify patients from the AMARANTH study (of BACE1 inhibitor, lanabecestat) into 2 groups: slow vs rapid progressors. They then analyzed the treatment effects on biomarker ($A\beta$) and clinical outcomes (CDR-SOB and ADAS-Cog13) with a mixed model for repeated measure for each PPM-stratified group. PPM-derived stratification provided a more sensitive tool for assessing treatment effects. It appears to show evidence for a slowdown of cognitive decline in the lanabecestat-treated group of slow-progressors, suggesting that the higher drug dose may slow down AD progression at earlier stages. The authors also showed that using PPM-guided stratification for clinical study design could significantly decrease sample size. In the AMARANTH trial, if it had been designed with only slow progressors (who showed the biggest cognitive benefit according to the authors' results), it could have cut down the sample size by 90%.

As there is a lot of heterogeneity in humans, it is critically important to stratify them properly into clinical trials to ensure we do not miss potential benefits of drugs that would otherwise be cast aside due to no benefit in the overall group. Thus, an approach like this could significantly help the field.

Thank you for clearly summarising our work and your positive comments, acknowledging the importance and novelty of our work.

There are some changes to the manuscript that are necessary to help make it clearer to the reader:

1. Supplemental Figures/Tables are not in the order they are referenced to throughout the manuscript. Thank you for spotting this. We have now fixed this in the revised manuscript.

2. Figure 3A: legend is cut off.

Thank you for spotting this. We have now revised this figure ensuring the legend appears within the figure space.

3. Please include tables summarizing the statistical treatment effects/significant interactions for $A\beta$, CDR-SOB and ADAS-Cog13 (pages 10, 11, 13 & 14).

Thank you for this suggestion. We have now included the suggested table in SI (Table S4)

4. Please indicate the significant findings in figures 3, 4 and SI with asterisks.

Thank you for this suggestion. We have included asterisks in the figures indicating significant results.

5. In the last paragraph on pg 14, mention the p-value for the ADAS-Cog13 and describe the MMRM interaction results as was done for CDR-SOB (or provide this in supplement as well as a table summarizing this in supplement).

Thank you for this suggestion. We have included more details in the revised text and tables on the ADAS-Cog13 analyses in SI (Table S4).

6. On p 16, the 2nd paragraph that describes the transitioning of individuals from one group to the other: please provide a graphical representation of this as well as a table that can be in supplement.

Thank you for this suggestion. We have included Figure 5, as suggested.

7. Fig 5: the description under the graph says that the 'all progressive' group is 'red' but the legend and the line on the graph for 'all progressive' is green.

Thank you for spotting this. We have now revised the figure and the caption to match.

Reviewer #4

Thank for supporting the review of our manuscript.

Response to reviewers

Reviewer #1 (Remarks to the Author):

REJECT: Manuscript does not presents novel contributions to be published in NCOMMs

We would like to take the opportunity to clarify the novelty of our approach. We have built a model architecture (PPM) that supports multimodal feature extraction, precise classification of patients to clinically stable vs. progressive and trajectory modelling to derive individual patient prognosis. We have previously validated our model predictions against multiple independent data samples and longitudinal real-world patient data.

The novelty of the work presented in this paper lies in demonstrating that our modelling approach can be successfully applied for patient stratification in clinical trials enhancing their efficiency and outcomes. It is important to note that our model was trained on research datasets (ADNI) and tested on completely independent and truly out-of-sample clinical trial data (AMARANTH), demonstrating the strong potential of our modeling approach for translation in clinical settings. In particular, we use the AMARANTH data as an independent test sample, stratify the patients at baseline into slow vs rapid progressive and demonstrate that patients in the slow progressive group show significant reduction in cognitive decline due to lanabecestat treatment. This result provides evidence that using our AI-guided tool to re-stratify patients rescues a clinical trial that was deemed futile; that is, if the right patients were included in the trial (i.e. slow progressive patients at earlier neurodegeneration stages) the trial would have met the primary outcome of reduction in cognitive decline. We believe that our modelling approach and findings make a novel and important contribution with strong translational impact for clinical trials and drug discovery.

Reviewer #2 (Remarks to the Author):

I am satisfied with the updated manuscript.

We are grateful to the reviewer for their positive and constructive feedback -it has allowed us to clarify our approach and strengthen our manuscript

Reviewer #3 (Remarks to the Author):

The manuscript has been significantly improved in this revision.

We thank the reviewer for their constructive feedback -it has helped improve our manuscript and strengthen our findings.

Two minor comments remain:

1. In the new Table S4, please denote the p-values that reach statistical significance with bold text for clarity.

We have revised the table following the reviewer's suggestion.

2. On the first line of p. 14, the text reads: "Similar analyses in the All Progressive group (n= 1354, including individuals in the slow and rapid progressive groups) showed significant increase rather than decrease in CDR-SOB scores compared to the placebo group." a. This statement refers to the right-most panel of Figure 4A & B. The "All Progressive" panels appear to show a decrease (rather than an increase) in CDR-SOB scores in the 50 mg group compared to placebo.

We thank the reviewer for spotting this inconsistency between the figure and the text. We apologise for the confusion. We have now revised the text and added more detailed statistical analysis to clarify the results presented in Figure 4, showing no significant differences in the All progressive group for CDR-SOB.

In particular, the text writes:

‘Similar MMRM analyses for the All Progressive group (n= 1354, including individuals in the slow and rapid progressive groups) showed significant main effects of Timepoint ($F(2,2195.0) = 23.68, p < 0.001$) and Treatment ($F(2,1883.7) = 3.68, p = 0.03$), but no significant Treatment x Timepoint interaction ($F(4, 2176.9) = 2.00, p = 0.09$). Further, we did not observe significant changes in CDR-SOB scores over time (week 104 minus week 1) for lanabecestat 20mg (Welch's Two Sample t-test, $t(266.93) = -0.35, p = 0.72$) nor lanabecestat 50mg (Welch's Two Sample t-test, $t(242.09) = 1.16, p = 0.25$) compared to placebo. These results are consistent with lack of significant slowing of cognitive decline due to treatment, as previously reported for the AMARANTH trial²¹.’

b. The text mentions this is a significant change, so please add the asterisk to the corresponding panel.

As explained above, there are no significant differences, as shown in the All Progressive plots in Figure 4.

Reviewer #4 (Remarks to the Author):

I co-reviewed this manuscript with one of the reviewers who provided the listed reports. This is part of the Nature Communications initiative to facilitate training in peer review and to provide appropriate recognition for Early Career Researchers who co-review manuscripts. We thank the reviewer for supporting the review of our manuscript.